# Language Embedded Radiance Fields for Zero-Shot Task-Oriented Grasping

**Adam Rashid\*, Satvik Sharma\*, Chung Min Kim, Justin Kerr,**
**Lawrence Yunliang Chen, Angjoo Kanazawa, Ken Goldberg**

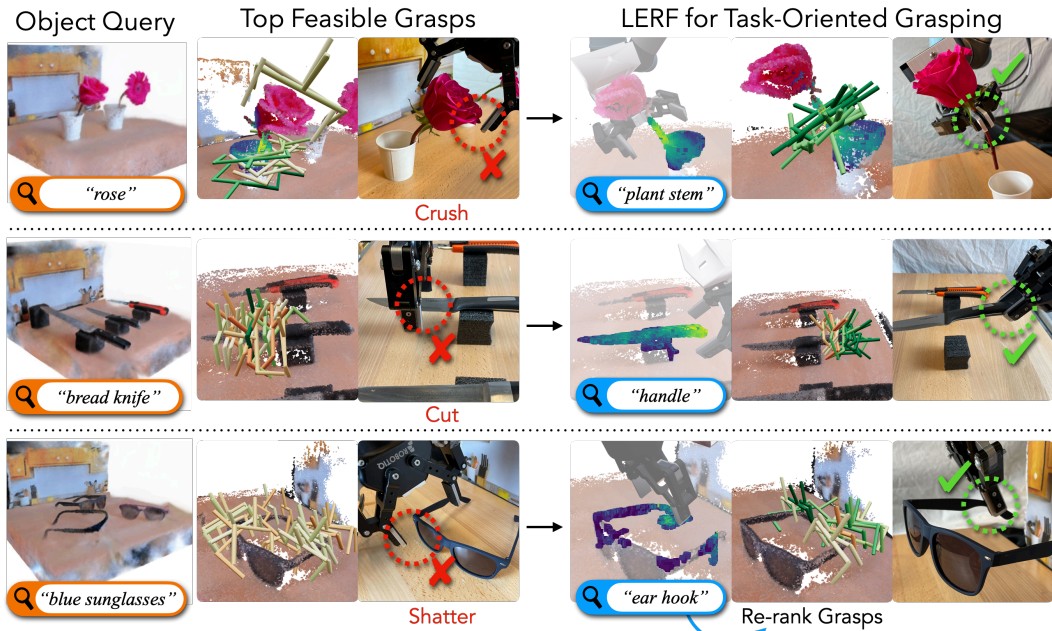

Figure 1: Learning-based grasp planners primarily consider object geometry, potentially yielding suboptimal grasps. LERF-TOGO uses natural language to select the target object with LERF [1] (in **orange**), and resamples grasps towards on object subparts using conditional LERF queries (in **blue**) for safe, task-oriented grasps.

**Abstract:** Grasping objects by a specific subpart is often crucial for safety and for executing downstream tasks. We propose LERF-TOGO, Language Embedded Radiance Fields for Task-Oriented Grasping of Objects, which uses vision-language models zero-shot to output a grasp distribution over an object given a natural language query. To accomplish this, we first construct a LERF of the scene, which distills CLIP embeddings into a multi-scale 3D language field queryable with text. However, LERF has no sense of object boundaries, so its relevancy outputs often return incomplete activations over an object which are insufficient for grasping. LERF-TOGO mitigates this lack of spatial grouping by extracting a 3D object mask via DINO features and then conditionally querying LERF on this mask to obtain a semantic distribution over the object to rank grasps from an off-the-shelf grasp planner. We evaluate LERF-TOGO's ability to grasp task-oriented object parts on 31 physical objects, and find it selects grasps on the correct part in 81% of trials and grasps successfully in 69%. Code, data, appendix, and details are available at: lerftogo.github.io

**Keywords:** NeRF, Grasping, Semantics, Natural Language

---

\* Denotes equal contribution, alphabetically ordered
UC Berkeley

7th Conference on Robot Learning (CoRL 2023), Atlanta, USA.

# 1 Introduction

Many common objects must be grasped appropriately to avoid damage or facilitate performing a task: a knife by its handle, a flower by its stem, or sunglasses by their frame. Learning-based grasping systems exhibit impressive robustness on grasping arbitrary objects [2, 3, 4, 5, 6, 7, 8, 9, 10], but these systems typically measure grasp success based on whether the object was lifted [11, 12, 13, 14, 15]. Critically, these methods ignore an object's semantic properties: even if a robot could locate your favorite sunglasses, rather than safely grasp at the frame it may shatter the lenses. This ability to grasp an object part based on a desired task and constraints is called *task-oriented grasping*, and while well-studied [16, 17, 18, 19, 20, 21], previous methods collect specific object affordance datasets and struggle to scale to a diverse set of objects. Instead, the flexibility of natural language has the potential for specifying what and where to grasp. In this work, we propose *LERF* for *Task-Oriented Grasping on Objects* (LERF-TOGO), a method which enables task-oriented grasping through natural language by using large vision-language models in a zero-shot manner.

LERF-TOGO takes as input an object and a task-orienteed object part name in natural language (i.e. *"flower; stem"*), and outputs a ranking over viable grasps on this object from which the robot should grasp. We build on recent work Language Embedded Radiance Fields (LERF) [1], which takes in calibrated RGB images and trains a standard NeRF in tandem with a scale-conditioned CLIP [22] feature field. Given a sentence prompt query, it outputs a 3D *relevancy* heatmap representing similarity to the query. However, these heatmaps may fail to highlight the full object (e.g., highlight only the bristles of a brush), which may cause issues when directly deployed to a task-oriented grasping task (grasp the "handle" of a brush). LERF-TOGO improves upon LERF's capabilities by predicting a 3D object mask using 3D DINO [23] features explicitly during inference. We propose a method of conditional LERF querying which restricts an object sub-part query to the object mask, leveraging the multi-scale nature of LERF to isolate specific regions within an object. LERF-TOGO then uses GraspNet [15] to generate grasps, re-ranking them based on the geometric and semantic distributions. We implemented a system with appropriate regularizations which allows LERF-TOGO to operate on a physical robot and evaluate on 39 common household objects. In experiments, 96% of grasps are on the correct object, 82% on the correct object part, and 69% result in a successful grasp.

This work contributes LERF-TOGO, an algorithm for producing task-based semantic grasp distributions over an object by first extracting a coarse 3D object mask with DINO [23] features to locally grow a relevant region, then conditioning a LERF query on this mask to isolate sub-parts of an object. We design a robotic system which integrates LERF-TOGO on a physical robot to reconstruct a LERF of a scene, then execute task-oriented grasps through natural language to grasp semantically meaningful object parts. See the project website at: lerftogo.github.io

# 2 Related Work

## 2.1 Task-Oriented Grasping

Task oriented grasping studies how to grasp objects by specific parts based on a use case. It has been studied by probabilistically modeling human grasps [19], extracting geometric features from labeled object parts [21], training on part-affordance datasets in simulation [24], or transferring category-specific part grasps to new instances [25]. Recent works [26, 17] train object-part grasp networks by leveraging object part and manipulation affordance datasets for a range of household objects. Kokic et al. [18] use videos of humans interacting with objects to guide grasps towards the same part. Decomposing objects into parts has also long been studied as a co-segmentation task in vision [27, 28]. Recent approaches use pretrained vision features to discover common parts within sets of objects [29]. This technique has been applied at scale to segment parts of objects based on a canonical object [30] or detect object affordances from example images of human usage [16]. Though effective, it assumes access to a canonical image of each object and pre-existing part labels or demonstrations, which are restrictive in real-world applications. In contrast, LERF-TOGO uses off-the-shelf vision-language models trained at scale, so it captures long-tail object and parts more easily without the use of affordance datasets.

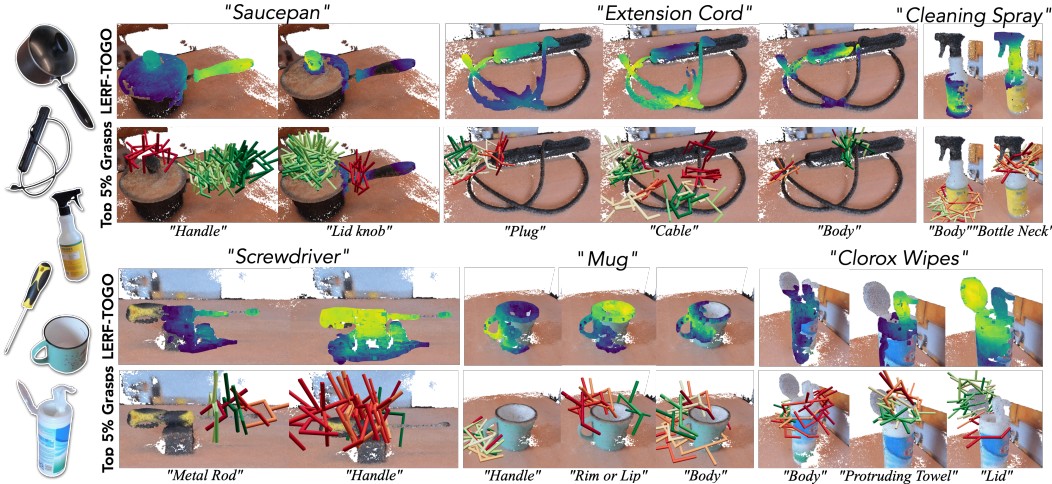

Figure 2: **Task-Oriented Grasps**: LERF-TOGO grasps target objects by different parts with different natural language queries (in quotes). Left: Crops of target objects. Right: Top row visualizes the relevancy distribution across the object mask, and bottom row shows top 5% of resampled grasps.

## 2.2 Neural Radiance Fields (NeRF)

Neural Radiance Fields (NeRF) [31] are an attractive representation for high quality scene reconstruction from pose RGB images, with an explosion of recent work on visual quality [32, 33, 34, 35, 36, 37, 38], large-scale scenes [39, 40, 41], optimization speed [42, 43, 44, 45], dynamic scenes [46, 47, 48], and more. Because of its high-quality reconstruction and differentiable properties, NeRF has been widely explored in robotics for navigation and mapping [32, 49, 50, 51], manipulation [52, 53, 54, 55], and for synthetic data generation [56]. This work is most similar to works such as Evo-NeRF [54] which use NeRF as a real-time scene reconstruction to grasp objects. However, in contrast to previous works which only use RGB information, in this work we must include additional semantic information in 3D to select grasps falling on relevant target objects.

Several prior works explore using semantic outputs inside NeRF. Semantic-NeRF [57], Panoptic Lifting [58], and Panoptic Neural Fields [59] distill semantic categories from semantic segmentation networks into 3D to improve the 3D consistency of labels, particularly noting the denoising effect of averaging multiple views. Other works such as Distilled Feature Fields [60] or Neural Feature Fusion Fields [61] distill feature vectors from DINO and LSeg [62], and show they can be used for editing and scene segmentation. We build off of LERF [1], which is described in the next section.

## 2.3 LERF Preliminaries

Language Embedded Radiance Fields (LERF) [1] is a recent representation that distills CLIP features into a NeRF. LERF inputs RGB images with camera poses and outputs a 3D field of DINO embeddings as well as a scale-conditioned CLIP field. This supports querying points in 3D for CLIP embeddings at different physical scales, capturing different semantics given different amounts of context. Given a text query, a relevancy value (from 0 to 1) can be generated at any 3D point by calculating the cosine similarity between LERF-queried embeddings and the CLIP embedding of query text. During this query process, a grid search on the scale parameter retrieves the scale with the highest activation. LERF is particularly attractive for task-oriented grasping because 1) its multi-scale parameterization allows queries at both object-level and part-level scales 2) LERF uses outputs from a pre-trained CLIP model without fine-tuning, which supports a variety of long-tail object queries not included in object or part segmentation datasets. LERF, however, tends to produce nonuniform activations on object queries because it lacks spatial grouping as shown in Fig. 5. In this work we show how to explicitly use the DINO feature field to obtain object masks to enable down-stream object part queries related to task-oriented grasping.

### 2.4 Natural Language in Robotics

With the advent of large pretrained language and vision-language models, several works have explored building 3D map representations to guide robot navigation. VL-Maps [63] and Open-Scene [64] build a 3D language embedding from pretrained open-vocab detectors [65, 62], which can be used to navigate to target queries. CLIP-Fields [66], ConceptFusion [67], and NLMaps-SayCan [68] take more of a region-proposal based approach, querying CLIP on the outputs of some region proposal methods and fusing them into 3D point clouds for downstream navigation tasks. Region-based zero-shot methods retain more language understanding than fine-tuned features but run the risk of missing objects by insufficiently masking input images. Semantic Abstraction [69] avoids this by extracting relevancy from vision-language models using Chefer et al. [70] and uses these for composing multiple language queries with spatial relationships. Language has also been studied in the context of robot manipulation. Mees and Burgard [71] use the MAttNet [72] vision-language model for object rearrangement, and CLIPort [73] uses language understanding from CLIP to train a language-conditioned pick and place module from demonstrations. PerAct [74] uses language-conditioned demonstrations with a 3D scene transformer to learn diverse tasks, MOO [75] uses the outputs from OWL-ViT to condition a manipulation policy for grasping objects, and large-scale demonstration datasets like RT-1 [76] train on massive language-conditioned demonstration trajectories. In contrast to many other language-conditioned approaches, LERF-TOGO uses internet-scale vision models purely zero-shot and does not require fine-tuning on demonstrations or robot exploration.

## 3 Problem and Assumptions

Given a planar surface (table or workbench) containing a set of objects, the objective is for a robot to grasp and lift a target object specified using natural language. This query (e.g., *"sunglasses; ear hooks."*) includes both the object query (*"sunglasses"*) and the object part query, which specifies the part to grasp the object by (*"ear hooks"*). We experiment with lifting this assumption in Sec. 5 by leveraging an LLM for providing part queries. We assume access to a robot manipulator with a parallel jaw gripper and calibrated wrist-mounted RGB camera, and the objects in the scene are graspable by the robot. We also assume the object query specifies a present single object.

## 4 Method

Given an object and object part query, LERF-TOGO outputs a ranking of viable grasps on the object part. To accomplish this, it first captures the scene and reconstructs a LERF (Details in Section B). Given a text query, LERF can generate a 3D relevancy map that highlights the relevant parts of the scene (Sec. 2.3). Second, a 3D object mask is generated using the LERF relevancy for the object query and DINO-based semantic grouping (Sec. 4.1). Third, a 3D part relevancy map is generated with a conditional LERF query over the object part query and the 3D object mask (Sec. 4.2). The part relevancy map is used to produce a semantic grasp distribution.

### 4.1 3D Object Extraction

An important limitation of LERF is its lack of spatial grouping within objects: for example given *"can opener"*, LERF tends to highlight regions of the object that most obviously identify that object (e.g. the metal cogs on the can opener as shown by the orange star in Fig. 3). However, since the region that visually identifies the object and the desired grasp location (e.g. handle) can differ significantly, this is problematic. LERF inherently exhibits such local behavior because it trains on local crops of input images, causing CLIP embeddings surrounding the handle to be unaware if it belongs to a can opener. LERF-TOGO overcomes this by finding a 3D object mask given a language query, which groups the object part together with the LERF activation. To create the object mask, we leverage the 3D DINO embeddings [23] (self-**DI**stillation with **NO** labels) present within LERF during inference, because DINO embeddings have been shown to exhibit strong object awareness and foreground-background distinction [29, 23, 77].

First, we obtain a coarse object localization from LERF by rendering a top-down view of the scene and querying the object. We produce a foreground mask by thresholding the first principal compo-

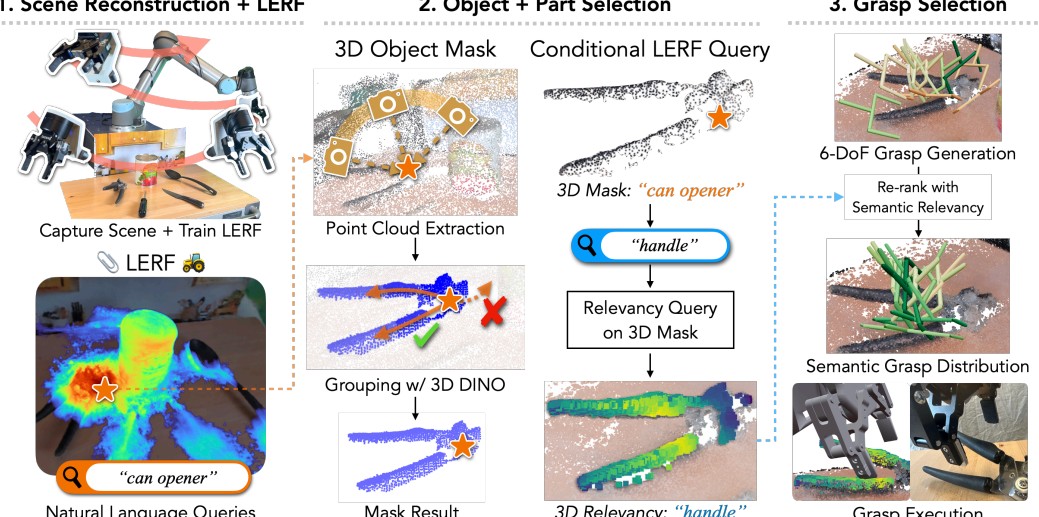

**1. Scene Reconstruction + LERF**

Capture Scene + Train LERF

📎 LERF 🚜

"can opener"

Natural Language Queries

**2. Object + Part Selection**

3D Object Mask

Point Cloud Extraction

Grouping w/ 3D DINO

Mask Result

Conditional LERF Query

*3D Mask: "can opener"*

"handle"

Relevancy Query on 3D Mask

*3D Relevancy: "handle"*

**3. Grasp Selection**

6-DoF Grasp Generation

Re-rank with Semantic Relevancy

Semantic Grasp Distribution

Grasp Execution

Figure 3: **LERF-TOGO Pipeline**: After reconstructing the scene with a wrist-mounted camera, we render an object-centric point cloud around the highest LERF activation. We next extract a 3D object mask by flood-filling the DINO features in this point cloud, condition an object part query on this object mask. Finally, we sample grasps and re-rank them according to 3D object part relevancy.

nent of the top-down rendered DINO embeddings, and constrain the relevancy query to this mask to find the most relevant 3D point. We then refine this single-point localization into a complete object mask. We render an object-centric point cloud around this 3D point by deprojecting NeRF depth from multiple views, and then iteratively grow the object mask by including neighboring points to the frontier which lie within a threshold DINO similarity (similar to floodfill). The output of this process is a set of 3D points lying on the target object. See the Appendix for more details.

### 4.2 Conditional LERF Queries

Another important challenge of using CLIP is its tendency to behave as a bag-of-words [22]: the activation for *"mug"* behaves very similarly to *"mug handle"* because CLIP latches onto individual words, not the grammatical structure of sentences. To mitigate this phenomenon, LERF-TOGO introduces a conditional method of querying LERF relevancy by composing two related queries, similarly to how composing prompts has shown promise in generative modeling for guiding specific properties [78]. Because LERF is scale-conditioned, during inference it searches over scales for a given query and returns the relevancy at the scale with the highest activation. To condition a LERF query, LERF-TOGO searches only on the points within the 3D object mask. Intuitively, this results in a distribution over the object's 3D geometry representing the likelihood that a given point is the desired object part, which can be used for biasing grasps towards this region.

### 4.3 Grasping

**Grasp Sampling** Ensuring complete coverage of grasps on objects is critical to avoid missing specific object parts. We use GraspNet [15], which can generate 6-DOF parallel jaw grasps from a monocular RGBD point cloud, but from a single view it often misses key grasps on target object parts. To mitigate this, and to leverage the full 3D geometry available within NeRF, we create a hemisphere of virtual cameras oriented towards the scene's center. For every virtual camera, we convert the scene's point cloud to the camera coordinate frames before providing it as input to the pretrained GraspNet model. To obtain the final set of grasps for the scene, we combine the generated grasps from the virtual cameras using non-maximum suppression to remove duplicates.

**Grasp Ranking** Given the grasps in the previous step (the *geometric* distribution), we now combine it with the semantic distribution across an object obtained from LERF-TOGO. The semantic score $s_{sem}$ for a given grasp is computed as the median LERF relevancy of points within the grasp volume. The geometric score $s_{geom}$ is the confidence output from GraspNet, indicating grasp quality based

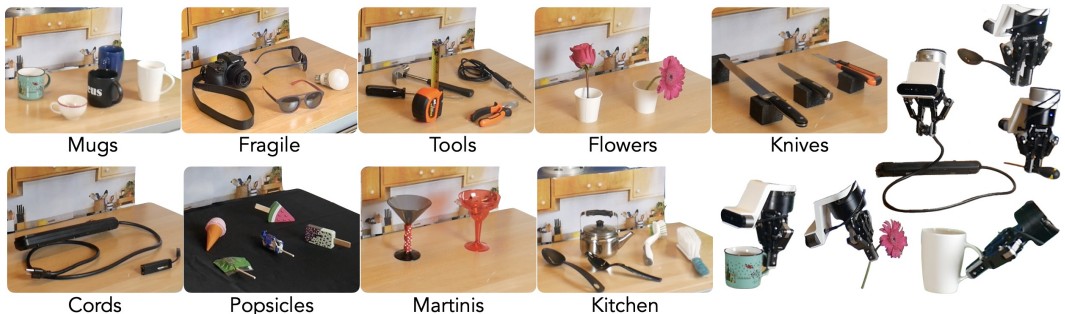

Mugs     Fragile     Tools     Flowers     Knives

Cords     Popsicles     Martinis     Kitchen

Figure 4: **Left**: Select scenes used during experiments, **Right**: example grasps using LERF-TOGO.

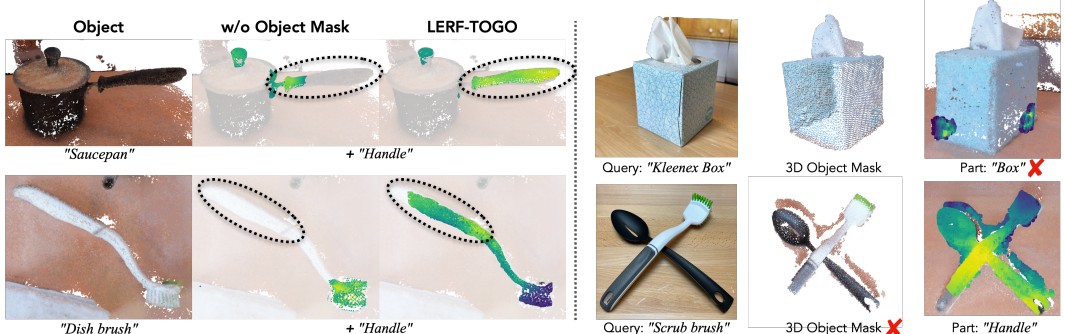

Figure 5: **Ablation and Limitations**: *Left*: Without 3D object masking and conditional querying, LERF cannot capture oblong object shapes. *Right*: CLIP can sometimes fail on generic prompt queries, like the poor activation on the box. Additionally, LERF-TOGO struggles with groups of connected objects as the 3D object mask groups them all together.

on geometric cues. To balance relevance and success likelihood, we combine the grasp score $s = 0.95s_{sem} + 0.05s_{geom}$ to ensure that we consider the most relevant grasps while slightly biasing towards confident grasps.

## 5 Experiments

**Part-Oriented Grasping** We evaluate LERF-TOGO on a wide variety of 31 different objects and 49 total object parts to grasp (Fig. 4). For each object, we select an object query by describing it sufficiently to unambiguously differentiate between other objects in the scene. We use semantic descriptions when possible, and add visual descriptions *only* when such descriptions are ambiguous (i.e using color to differentiate multiple mugs in a scene). We provide a part query for each object by describing a natural place for a robot to grasp and lift (i.e. "handle", "plant stem", "ear hook", "frame"). In addition, several objects include different grasp locations. A grasp is successful if it lifts the correct object using the appropriate subpart at least 10cm vertically, and the object remains securely within the gripper jaws throughout. For each query, we measure 1) whether the selected grasp was on the correct *object*, 2) whether the selected grasp was on the correct object *part*, and 3) whether the grasp successfuly *lifted* the object from the table. Every scene is reconstructed once in the beginning, after which the objects are removed sequentially (i.e., objects are removed one-by-one) with no updates in the scene representation. For a full list of object queries see the Appendix.

|  | ConceptFusion [79] | LERF-TOGO |
|---|---|---|
| Correct Object | 77% | **96%** |
| Correct Part | 39% | **82%** |
| Successfuly Lifted | – | **69%** |

Table 1: **Part-Based Grasping Results:** Results are reported across 49 different prompts and 12 scenes. See the Appendix for a complete list of scenes and queries.

**Task-Oriented Grasping** LERF-TOGO accepts a natural language part query as input, allowing it to be used alongside large language models (LLMs) to generate parts based on the task. To investigate if the LLM can also generate the object part, we use an LLM (ChatGPT) to generate the object and part query automatically via few-shot prompting. Results are shown in Table 3. The prompt and all tasks are included in the Appendix. Given the task and the list of objects in the scene, the LLM is tasked with generating the correct object and object part pair (object, part). We used a majority voting scheme to query the LLM. Given the task, the LLM provides seven candidates that we use to select the pair (object, part) that appears in a majority of the responses. We also mention details to integrate with an LLM planner in Section D.3 in the Appendix.

## 5.1 Comparisons

|                | SemAbs [77] | OWL-ViT [69] |
|----------------|-------------|--------------|
| Correct Object | 80%         | **85%**      |
| Correct Part   | 35%         | **50%**      |

Table 2: **Single View Comparisons:** Results are reported across 20 different prompts and 5 scenes. See the Appendix for a complete list of scenes and queries.

|                | Human-Query | LLM-Query |
|----------------|-------------|-----------|
| Correct Object | **96%**     | 96%       |
| Correct Part   | **82%**     | 71%       |

Table 3: **Task-Oriented Success:** Results using an LLM to choose the object part given a task specification. Results are reported across 49 different prompts and 12 scenes.

**ConceptFusion** [79] generates a multimodal point cloud of a scene by fusing RGBD images and their extracted features together. To query ConceptFusion, we provide it with the concatenated object and part prompt (i.e. *"mug handle"*) and rank grasps via the highest similarity. We report the object and part success without physical evaluation. More details are in Appendix Section D.2.

**Semantic Abstraction** [69] takes a single RGBD frame as input and a text query and outputs a relevancy heat map over the image. A query is a success if the majority of the heatmap overlaps with the object part. In instances where activations are detected on other objects, it is considered successful if the highest activation is on the desired object part. Detailed results can be found in Table 2 and the Appendix.

**OWL-ViT** [80] is an open-vocab detector which takes in an RGB image and text prompts and outputs segmentation maps. We provide OWL-ViT a single input image that encompasses all object parts for a fair comparison. To obtain an object mask, we use the object prompt to establish an initial bounding box. This box serves as a region to identify the highest-scoring part within the region. In order to deem the part box as successful, we visually confirm that it aligns with the object part. Results can be found in Table 2 and the Appendix.

## 6 Results

**Part-Oriented** LERF-TOGO overall achieves a 69% success rate for physically grasping and lifting objects by the correct part. The selected grasp was located around the correct object part 82% of the time, with the remaining failures being grasp execution failures. For context, the highest confidence geometric grasp on an object mask only lies on the correct part 18% of the time, suggesting LERF-TOGO meaningfully biases the grasp distribution to the object part. Selected task-oriented queries are visualized in Fig. 2: the distribution of grasps drastically shifts based on the given part query, and can focus task-oriented grasps on multiple different regions per object based on the language prompt. LERF-TOGO shows strong language understanding performance for object selection (96%), able to differentiate between very fine-grained language queries like color, appearance ("matte" vs "shiny"), or semantically similar categories (*"popsicle"* vs *"lollipop"*). It also can recognize long-tail object queries like *"ethernet dongle"*, *"cork"*, or *"martini glass"*, owing to its usage of CLIP zero-shot.

**Task-Oriented** Combining few-shot LLM prompting with LERF-TOGO identifies the correct primitive with 92% success and produces grasps on the correct object with 71% success across 49 tasks on 39 different objects. The LLM could identify the object in the scene with the same success rate as the human, giving the correct object and part pairs for tasks like "scrub the dishes" and "cut the

steak". However, the LLM had a lower success rate (71%) compared to the human (82%) for object part selection. This is because CLIP, and by extension LERF-TOGO, can be sensitive to subtle variations in wording like "body" vs. "base" resulting in different LERF activations and thus grasps.

**ConceptFusion** LERF-TOGO out-performs ConceptFusion by 43% at task-oriented grasping because it can capture multi-scale semantics, while ConceptFusion is limited to one CLIP embedding per point. This makes hierarchical querying difficult, and is reflected by the fact that ConceptFusion performs similarly to LERF-TOGO at selecting the correct object, but suffers at selecting the right object part. Due to its lack of scale-conditioning, ConceptFusion frequently emphasizes sections of the table due to the inclusion of both the objects and the table itself in the mask proposals (Fig. 9).

**Semantic Abstraction** achieves an overall object detection rate of 80% and part detection of 35%. The method tends to produce empty relevancy responses when queried for specific object parts, potentially owing to its averaging across multiple scales which drowns out smaller part features. When presented with the object and part, the method highlights all of the object, owing to CLIP's bag of words behavior, a characteristic addressed by LERF-TOGO's compositional queries. (Fig. D.4).

**OWL-ViT** achieves 85% accuracy for object localization, struggling on very long-tail objects that were not encountered within the detection datasets. This behavior is amplified for object part queries, where queries tend to be long-tail such as *"measuring tape"* and *"ethernet dongle"* (Fig. 12).

**Failures** The primary failure modes of LERF-TOGO are mistaking visually similar object parts for one another (eg teapot spout for a handle), missing subtle geometries like the small teacup handle or spray trigger, or confusing very close categories like steak and bread knives. We also observe prompt sensitivity with part queries: for example *"bottle neck"* more strongly localizes grasps than *"neck"*, and without more prompt tuning *"body"* sometimes fails to highlight the bases of bottles.

**Object Extraction Ablation**: Without 3D object masking and conditional querying, LERF-TOGO suffers with oblong objects, as shown in Fig. 5. We compare against querying LERF individually for the object and part, and multiplying their results together. This produces fragmented results which can ignore relevant parts of the object for part queries.

# 7 Limitations and Future Work

One limitation of LERF-TOGO is speed: the entire end-to-end the process takes a few minutes which can be impractical for time-sensitive applications. Future work on additional regularizations and optimizations to LERF training may reduce computation time. Another key limitation of LERF-TOGO is with groups of connected foreground objects, as the DINO flood-fill mask will include the whole foreground group instead of isolating the individual components. Supporting hierarchy within foreground groups is critical to enable such cases. If the object query matches multiple objects in the scene (identical objects, or generic query like "mug" in the mugs scene), the system will arbitrarily choose only one of them. LERF-TOGO does not consider referring/comparative expressions (e.g., "mug next to the plate", "biggest mug"). In addition, though we present a method for obtaining object part queries from input task descriptions via LLMs, in future work we will evaluate its performance on a diverse set of tasks.

# 8 Conclusion

This paper presents LERF-TOGO, a method for using vision-language models zero-shot with Language Embedded Radiance Fields to grasp objects and their parts via language. By improving the spatial grouping of LERF relevancy outputs, LERF-TOGO can support hierarchical part queries conditioned on the full object. Results indicate it performs strongly at language-guided grasping, with grasps landing on the correct object 96% of the time, and furthermore can direct grasps to the correct object parts 81% of the time.

## Acknowledgements

This research was performed at the AUTOLAB at UC Berkeley in affiliation with the Berkeley AI Research (BAIR) Lab. The authors were supported in part by donations from Toyota Research Institute, Bosch, Google, Siemens, and Autodesk. Chung Min, Justin, and Lawrence are supported in part by the National Science Foundation Graduate Research Fellowship Program. Any opinions, findings, and conclusions, or recommendations expressed in this material are those of the authors and do not necessarily reflect the views of the Sponsors. We would like to thank our colleague Brent Yi for his work on Viser, the visualization tool of our experimental setup. We thank our colleagues who provided helpful feedback and suggestions, in particular Simeon Adebola and Julia Isaac.

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

# A  Appendix

## A.1  Implementation Details

We implement LERF-TOGO on top of the Nerfacto method from Nerfstudio [39]. For faster convergence and smoother optimization, we modify several parameters from the original LERF paper. We use a smaller hashgrid with 16 levels and a maximum resolution of 256, and find that using larger MLPs for the density, color, and transient output heads of NeRF results in faster convergence and better ability to handle specularities and robot shadows. We introduce weight decay of 1e-7 to the LERF network which smooths training. In addition, we compress the DINO embeddings into 128 dimensions and supervise on these vectors rather than the original DINO outputs, but we do not normalize the resulting vectors like Tschernezki et al. [61]. While constructing the CLIP embedding pyramid for LERF, we use crops ranging from 5% to 35% of image height with 6 pyramid levels, biasing the pyramid to smaller crops as LERF-TOGO is primarily interested in object and part queries.

# B  Robot Capture

**Scene Reconstruction** The robot uses a wrist-mounted camera to capture the scene with a hemispherical trajectory centered at the workspace, similar to Evo-NeRF [54]. The capture has a radius of 45 cm and arcs from $\pm100°$ around the workspace horizontally and an inclination range of $30°$ to $75°$. We capture images while the arm moves at 15 cm/sec at a rate of 3 hz, resulting in around 60 images per capture. We discard blurry images by analyzing the variance of the image Laplacian, ensuring the images are high quality. While the robot moves, we pre-process each image to extract DINO features, multi-scale CLIP, and ZoeDepth [81], which are used during LERF training. See Appendix B for details.

**Robot Capture Region Size** The robot captures the scene along a hemispherical trajectory arcing $\pm100°$ around the workspace horizontally ("1/2 hemisphere" in Fig. B.) When this horizontal sweep angle is reduced to a fraction of the range, the quality of the 3D object mask degrades, sometimes selecting the incorrect object altogether. LERF's semantic field is supervised on features of the scene images, thus the quality is heavily correlated with the distribution of images viewing the object. This lowers the quality of the 3D DINO embeddings used for the mask generation.

**LERF Training Steps** In our experiments the LERF scene representation is trained to 2k steps. As shown in Fig. B, objects/object parts (e.g., "spray nozzle", "bottle") can be detected in steps as low as 1k, but more fine-grained or smaller parts (e.g., "handle") may take longer (2-3k steps). This is consistent with what LERF reports: "fine-grained features take more steps [to emerge]".

**3D Object Extraction** Given the initial 3D point with the highest LERF activation, we create an object-centric point cloud by rendering six different views looking at the 3D coordinate. The views are $\pm90°$ around the upwards vector through the 3D point. For DINO floodfill, the threshold DINO similarity is defined as first projecting the current DINO embedding onto the first PCA component of the top-down image, then taking the L2 norm of the difference between the current embedding and the DINO embedding at the initial 3D point.

**NeRF Regularization** NeRF encounters difficulties in reconstructing texture-less planar surfaces, especially in the presence of specularities. This limitation is prominent in our table-top scenes, where the glossy surface and metallic objects can result in depth renderings with jagged missing regions. These missing regions can cause LERF renderings to spuriously activate and degrade the performance of grasp networks, so we apply depth regularization to mitigate this issue. We adopt the local depth ranking loss proposed in SparseNeRF [82] and use ZoeDepth [81] as the underlying depth model. We found this performs better than smoothness priors [83, 45] because it retains more fine-grained geometry. Additionally, we use the gradient scaling approach from Philip and Deschaintre [38], which significantly reduces the number of near-camera floaters and enables more robust grasping directly from point clouds rendered from the NeRF.

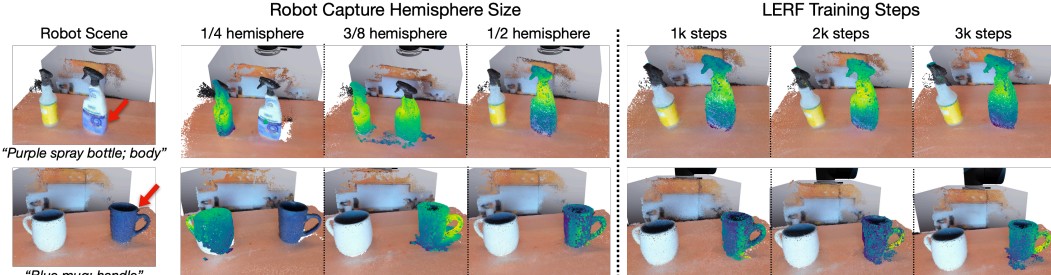

Figure 6: Left: Decreasing the robot scan region degrades the quality of 3D relevancy map generated by LERF-TOGO. Right: LERF-TOGO relevancy map converges earlier for larger objects and parts.

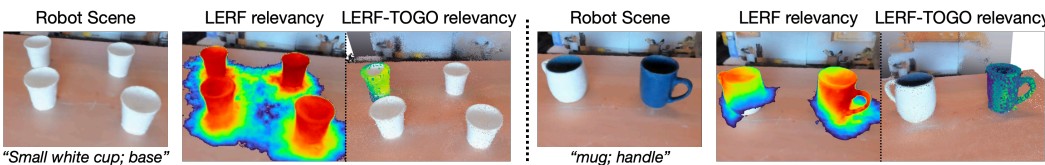

Figure 7: In the case of identical objects or ambiguous object queries, LERF-TOGO picks a single object in the scene but does not propose grasps for all the objects in the category.

Poses obtained from cameras in motion are slightly inaccurate, which we found could result in oversmoothed geometry with depth regularization. To overcome this, we optimize the NeRF for the first 500 steps without any regularization to allow the camera poses to settle, then anneal the depth regularization loss term from 0 to 100% over the next 1500 steps. Interestingly, we find staged training not only preserves thin features better but also speeds LERF optimization. We hypothesize this is because supervising the language field on un-converged density in free space results in a poor network initialization, while beginning LERF optimization after geometry has been largely removed from free space allows a smoother learning signal.

## C  Grasping

**Point Cloud Extraction** To extract a scene-wide point cloud for grasping, we use the method in Nerfstudio [39], which deprojects randomly sampled rays' depth from the train camera poses, then filters with outlier rejection. We then crop the point cloud to the workspace of the robot. For object centric point clouds, we deproject depth from views radially surrounding the object of interest.

**Motion Planning** A grasp is considered feasible if the robot can perform a collision-free trajectory with the following poses: the pre-grasp, grasp, and post-grasp configurations. The pre-grasp pose is positioned 5cm along the z-axis of the robot end effector, which allows the gripper to approach the target grasp pose with minimal additional motion. The post-grasp pose is located 10cm above the grasp pose, along the z-axis of the world frame. The UR5's IK solver calculates the set of viable joint configurations at these poses, and we calculate the trajectory as a linear interpolation between them. We additionally allow for a 180 degree rotation at the last wrist joint, as parallel-jaw grasps are rotationally symmetric. This facilitates the motion planning process, as the robot's camera mount is prone to colliding with the robot arm.

## D  Experiments

### D.1  Setup Details

We use a UR5 arm with a Logitech BRIO webcam at 1600x896 resolution, with all camera settings frozen before each capture prevent color discrepancies among images. The camera mount points

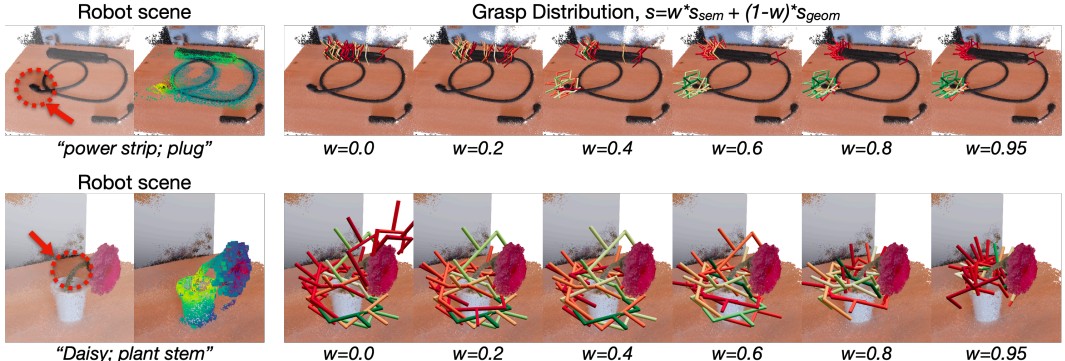

Figure 8: **Grasp score weighting**: Varying weight between geometric grasp score and semantic grasp score shifts the grasp distribution. A high semantic grasp weight ($w = 0.95$) is required, since geometric grasps may be biased away from small and fine-grained object parts of interest. Both geometric and semantic scores are in the range $[0, 1]$.

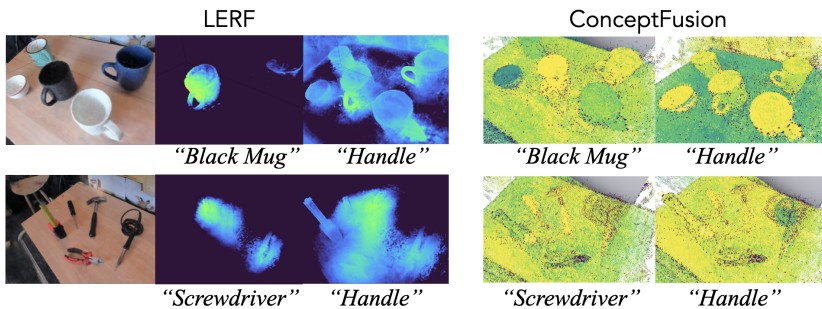

Figure 9: **Comparison with LERF and ConceptFusion**: ConceptFusion performs well on object-level queries, but struggles with sub-object part queries because of its lack of multi-scale semantics.

orthogonally to the gripper axis, to maximize the reachable workspace of the camera while pointing towards the workspace center. During robot capture, pre-computation of DINO, CLIP, and ZoeDepth is parallelized across 3 NVIDIA 4090 GPUs to achieve real-time performance, and all subsequent operations are carried out on a single 4090. Capturing a scene takes 30 seconds, training the LERF to 2k iterations takes 78 seconds, and finally querying LERF-TOGO takes 10 seconds.

### D.2 Comparison to Baselines

**ConceptFusion** We provide ConceptFusion with depth generated from the NeRF, which results in high-quality point clouds for grasping. To represent the paper, we use the OpenCLIP ViT-H/14 model, which is larger than the ViT-B/16 model for LERF-TOGO.

**Semantic Abstraction** Since the method takes a single image, we provide the method with an input image observing all object parts for a fair comparison. We provide it with part queries 2 ways and take the best performance: 1) the concatenated object and part prompt (i.e. *"mug handle"*), and 2) the object and part as separate queries.

### D.3 Integration with an LLM Planner

LERF-TOGO can integrate as a module with an LLM planner to combine task-oriented grasps for robotic manipulation tasks. We define a set of robotic manipulation primitives (grasp, press, twist, pick&place, pour) and prompt the LLM to output the correct primitive for a given task. We use the same majority voting scheme in the previous section to select both the correct robotic primitive and the pair (object, part). Now, given a task (e.g. 'uncork the wine'), an LLM can specify the action

| Scene | Object Query ; Part Query |
|---|---|
| Kitchen | (black matte spoon, handle), (shiny black spoon, handle), (teapot, handle), (dish scrub brush, handle), (dust brush, handle) |
| Flowers | (daisy, plant stem), (rose, plant stem) |
| Mugs | (blue mug, handle), (pink teacup, handle), (turqouise mug, handle), (white mug, handle), (black mug, handle) |
| Tools | (Measuring tape, base), (screwdriver, handle), (wire cutters, handle) (soldering iron, handle), (hammer, handle) |
| Knives | (bread knife, handle), (steak knife, handle), (box cutter, handle) |
| Martinis | (red martini glass, stem), (grey martini glass, stem) |
| Fragile | (camera, strap), (pink sunglasses, earhooks) (blue sunglasses, earhooks), (lightbulb, screw) |
| Cords | (power strip, plug), (power strip, base), (ethernet dongle, usb) (ethernet dongle, ethernet) |
| Messy | (ice cream, cone), (green lollipop, stick), (blue lollipop, stick) |
| Pasta | (wine, cork), (wine, bottle neck), (saucepan, lid knob) |
| Cleaning | (saucepan, handle), (corkscrew, handle) (clorox, wet towel), (clorox, lid), (clorox, body) (tissue box, box), (tissue box, tissue) |
| Bottles | (meyer's cleaning spray, spray trigger), (meyer's cleaning spray, bottle neck) (meyer's cleaning spray, body) (purple cleaning spray, body) (purple cleaning spray, spray trigger), (purple cleaning spray, bottle neck) |
| Electronics | (e-stop, red button), (computer mouse, scroll wheel), (controller, buttons), (spacemouse, button), (white controller; button) (controller, joystick), (e-stop, red button) |

Table 4: **Complete list of object and part queries**

to accomplish the task ('grasp') and the pair of object and object part (e.g. 'wine' and 'cork'). The prompt and all tasks are included in the Appendix.

### D.4   LLM Interface

We provide the full prompt to the LLM below. For any given task and scene the OBJECT_LIST is replaced with a list of objects within the scene and TASK is replaced with the desired task:

```
    Answer the question as if you are a robot with a parallel jaw gripper that has
    access to only the objects in the object list. Follow the exact format.
    First line should describe what basic action is needed to do the task from
    the following set of actions: press, grasp, twist, pick & place.
    The second line should only be an object from the object list followed by 1 object
    part that the robot would touch to do this task. VERY IMPORTANT: If the basic
    action is pick & place, only then have a third line with 'Place: '
    to specify the object to place on. \
Object list: ['pot', 'knife', 'spoon', 'black pan'] \
Q: How can I safely pick up a pan? \
Basic Action: grasp \
Sequence: 1. black pan 2. handle \
\
Object list: ['mechanical keyboard', 'knife', 'TV', 'camera'] \
Q: How can I safely hit the spacebar on a keyboard? \
```

```
Basic action: press \
Sequence: 1. mechanical keyboard 2. spacebar\
  \
Object list: ['green mug', 'blue spoon', 'fork', 'knife'] \
Q: How can I cut a block of cheese? \
Basic action: grasp \
Sequence: 1. knife 2. handle \
  \
Object list: ['salt shaker', 'knife', 'fork', 'white pan'] \
Q: How can I safely lift a salt shaker? \
Basic Action: grasp \
Sequence: 1. salt shaker 2. base \
\
Object list: ['red cup', 'blue cup', 'mug', 'bowl'] \
Q: How do I stack the red cup on the blue cup? \
Basic action: pick & place \
Sequence: 1. red cup 2. rim \
Place: blue cup \
  \
Object list: ['door knob', 'black mug', 'green dish brush', 'shiny knife'] \
Q: How do I open a door knob? \
Basic action: twist \
Sequence: 1. door knob 2. rim \
  \
Object list: ['dryer', 'washing machine', 'sunglasses'] \
Q: How do I turn on the washing machine? \
Basic action: twist \
Sequence: 1. washing machine 2. dial \
  \
Object list: ['paper towel roll', 'mug', 'teacup', 'headphones', 'pen'] \
Q: How do I grab a paper towel? \
Basic action: grasp \
Sequence: 1. paper towel roll 2. paper towel \
  \
Object list: ['magnifying glass', 'blue spoon', 'fork', 'knife'] \
Q: How do I pick up a magnifying glass? \
Basic action: grasp \
Sequence: 1. magnifying glass 2. handle \
  \
Object list: ['teddy bear', 'toy block', 'mouse', 'saucepan', 'hammer'] \
Q: How do I grab a teddy bear? \
Basic action: grasp \
Sequence: 1. teddy bear 2. head \
\
Object list: ['green mug', 'blue spoon', 'fork', 'knife'] \
Q: How do I put the mug in the cabinet? \
Basic action: pick & place \
Sequence: 1. green mug 2. handle \
Place: cabinet \
\
Object list: {OBJECT_LIST} \
Q: How can I safely {TASK}? \
Basic action: "
```

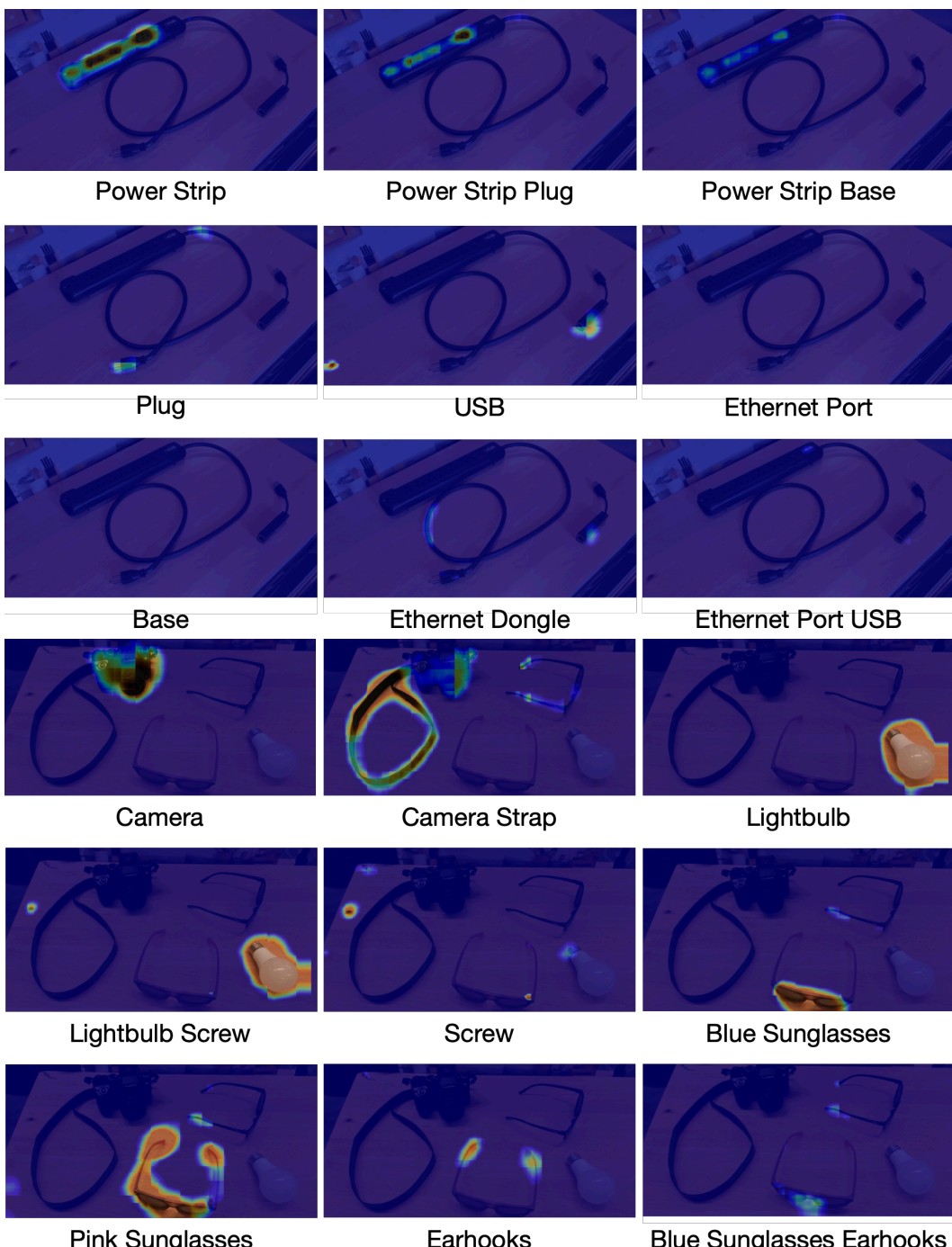

Figure 10: Semantic Abstraction results for object and object part localization (cont.)

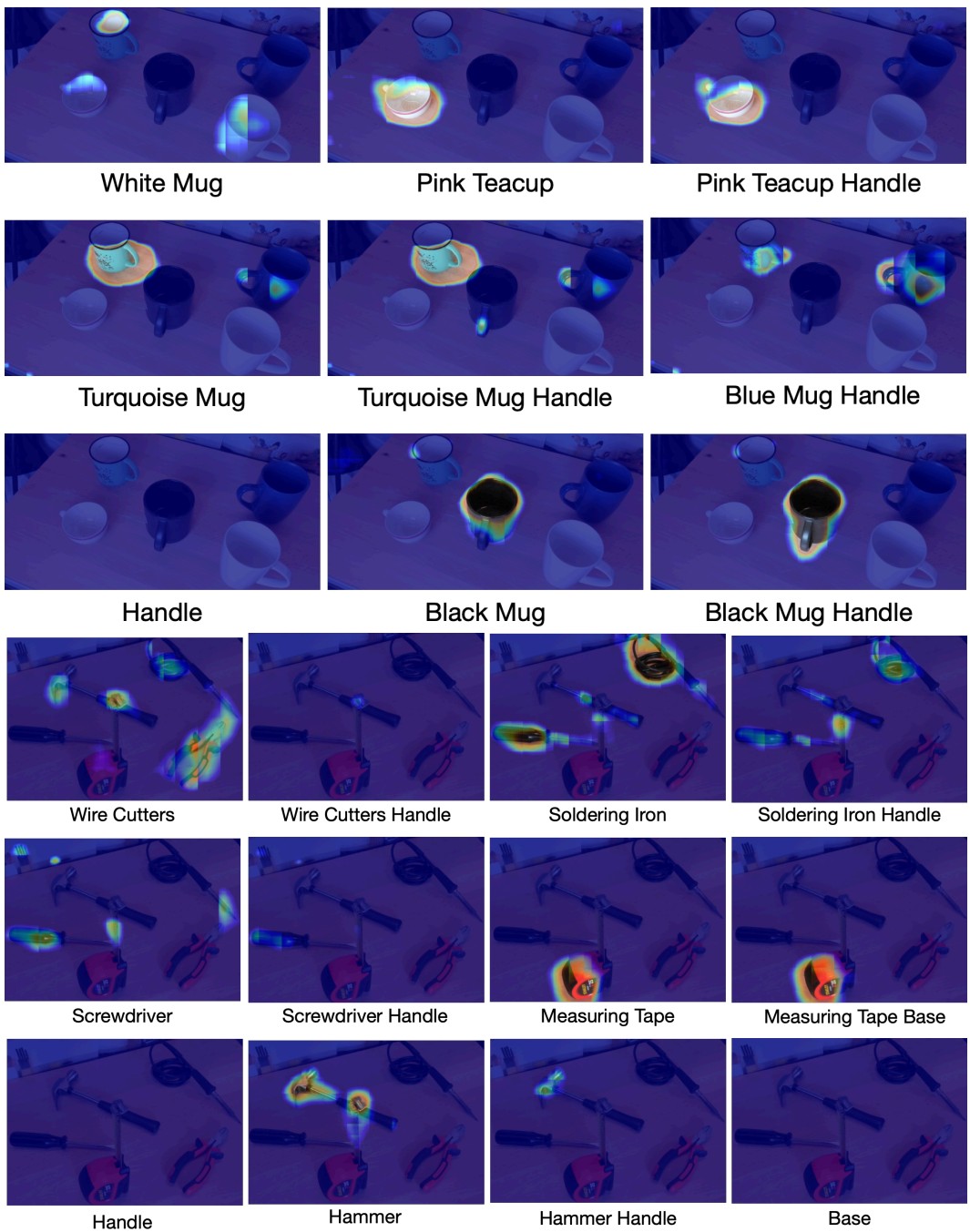

Figure 11: Semantic Abstraction results for object and object part localization

| Scene | Tasks |
|---|---|
| Kitchen | 'grab the matte spoon', 'pass the shiny spoon', 'grab the teapot', 'scrub the dishes', 'dust the book' |
| Flowers | 'give the daisy', 'give the rose' |
| Mugs | 'grab the blue mug', 'grab the pink teacup', 'grab the turquoise mug' 'grab the white mug', 'grab the black mug' |
| Tools | 'grab the measuring tape', 'give the screwdriver', 'cut the wire' 'hold the soldering iron', 'swing the hammer' |
| Knives | 'cut the bread', 'cut the steak', 'cut the box' |
| Martinis | 'deliver the grey martini glass', 'deliver the red martini glass' |
| Fragile | 'take a picture', 'wear the pink sunglasses' 'wear the blue sunglasses', 'move the lightbulb' |
| Cords | 'grab the power strip', 'plug in the power strip', 'plug the ethernet into the ethernet dongle', 'plug in the usb into the ethernet port' |
| Messy | 'eat the ice cream', 'eat the green lollipop', 'eat the blue lollipop' |
| Pasta | 'uncork the wine', 'grab the saucepan', 'open the saucepan', 'grab the corkscrew', 'lift the wine' |
| Cleaning | 'grab a wet towel from the clorox', 'close the box of clorox' 'hold the clorox', 'get a tissue', 'hold the tissue box' |
| Bottles | 'grab the meyers cleaning spray', 'open the meyers cleaning spray', 'spray the meyers cleaning spray', 'grab the purple cleaning spray', 'open the purple cleaning spray', 'spray the purple cleaning spray' |
| Electronics | 'hold down the e-stop', 'hit the scroll wheel', 'click the controller', 'click on that website with the spacemouse', 'click the white controller', 'turn video game character', 'release the e-stop' |

Table 5: **Complete list of tasks for each scene**

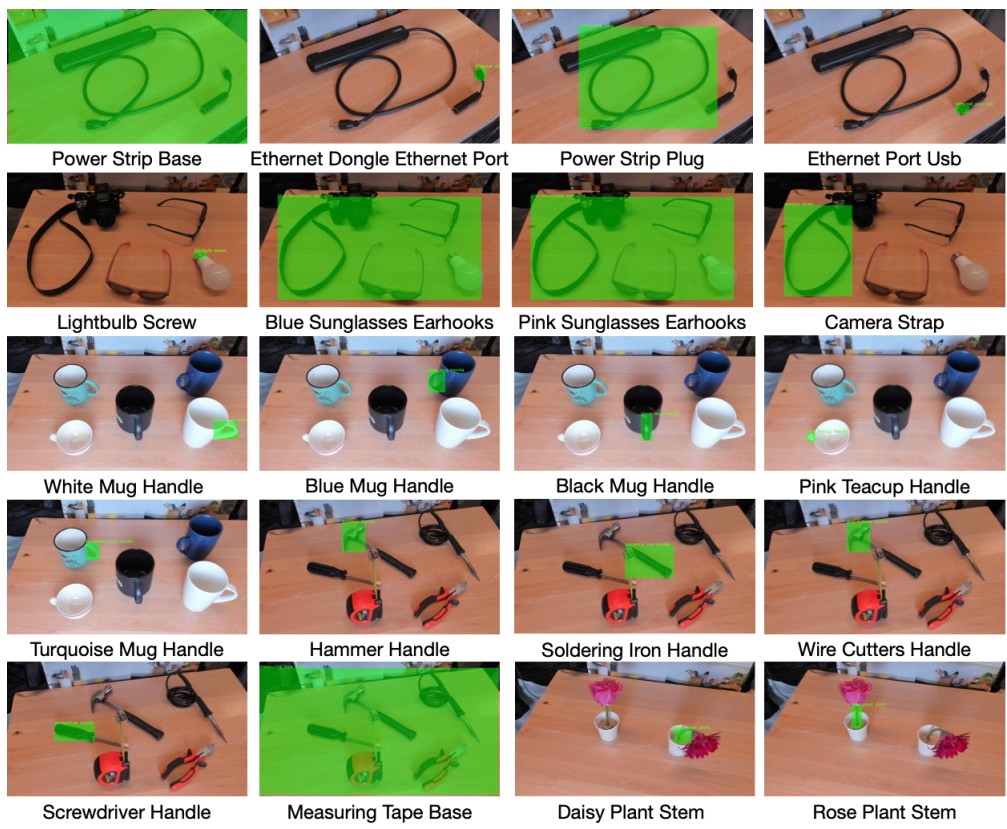

Figure 12: OWL-ViT results for object and object part localization

