# OpenReview forum: "Language Embedded Radiance Fields for Zero-Shot Task-Oriented Grasping"
_robot-learning.org/CoRL/2023/Conference — CoRL 2023 Oral_

### Official Review · Reviewer_aACe · 2023-07-17

**Confidence:** 4
**Originality:** Fair
**Technical Quality:** Good
**Clarity Of Presentation:** Fair
**Impact:** 3

**Recommendation:**

Weak Accept: I recommend accepting the paper, but will not argue for my recommendation if the majority of other reviewers have a different opinion.

**Review:**

**Summary of strengths and weaknesses**

Strengths:

- The demonstrated ability to perform "part-specific" grasping is useful
- Additionally, showcasing one way LERF can be used for representing + interacting with scenes in a semantic fashion for a robotics application is also a valuable contribution
- The general approach is intuitive and easy to digest, most likely making this a component that can be built upon and/or integrated into more full-stack systems.
- The paper includes good demonstrations of real-world capability

Weaknesses:

- The main weakness is the presentation of the method. While the high-level intuition can be understood pretty easily, I greatly struggled to understand many core details of how the pipeline actually works. It’s a major problem that there are absolutely no preliminaries on LERF included in the paper. The explanation of using the 3D-distilled DINO features for obtaining the object mask was very unclear.
- A second weakness is the mismatch between the framing of the paper, as *task*-oriented grasping, whereas in my view, the results shown only go so far as *part*-oriented grasping. It might sound pedantic, but these are not the same. Open-set part segmentation could have been a baseline you used. It’s both more non-trivial, and also potentially more valuable/interesting to provide actual “functional/long-term/downstream” task specifications and see what this could have done.
- Ablation evaluation - besides the full pipeline and integrating with grasping, the main innovation/contribution is the modifications to allow the part specification to work. But there are probably lots of other ways this could have been done, and showing at least some more exhaustive comparison (ideally quantitative) would be very beneficial
    - For example, the section *Object Extraction Ablation* says: *Without 3D object masking and conditional querying, LERF-TOGO suffers with oblong objects, as shown in Fig. 5. We compare against querying LERF individually for the object and part, and multiplying their results together. This produces fragmented results which can ignore relevant parts of the object for part queries —*. Ideally, there would be more of this kind of evaluation and analysis, since the specific design choices that allow part-specific queries are the main technical innovation (besides the general integration of LERF with a grasp predictor).

**Details**

*Technical Quality -*: Good. Strengths: The paper uses both new and (now-)standard tools in a simple but effective combination. The authors investigated and reported on key technical issues that arise when attempting this pipeline in a naive fashion, and include other details for running the whole system (compute, pre-processing, reconstruction, etc.). Weaknesses: The evaluations and ablations could have been stronger for supporting the claim that this is a good way to enable (i) LERF to perform well with part-specific queries and (ii) LERF relevancy maps to be combined with a grasp prediction model.

*Clarity -*: Fair. Strengths: The general problem setup and purpose of the paper are clear. Weaknesses: The clarity of the whole method was quite low. It was very hard for me to understand a significant portion of how the pipeline actually works and the core changes that were needed to allow LERF relevancy maps to be used properly.

*Originality -*: Fair. Strengths: The changes needed to make LERF work with part queries appeared to be important for the pipeline to operate. Weaknesses: The components in the pipeline are not new (LERF with 3D-distilled CLIP features and DINO features, 6-DoF graspnet). The idea of computing the relevancy map “hierarchically” (using only the coarsely segmented object to find the required object part), while being important to enable the robotics application, is also not a very significant innovation.

*Impact/Significance -*: Fair. Strengths: Part-specific grasping from language is likely to be useful in full-stack manipulation pipelines. Weaknesses: There are lots of 3D-distilled feature field/semantic point cloud pipelines being proposed, and there are lots of ways this overall pipeline could have been designed. From the evaluations and demonstrations, it's hard to walk away from this paper knowing "this is the right way to do it". Furthermore, the paper claims to be solving “task-oriented grasping” but in my view, it’s really addressing “part-oriented grasping”, which (while valuable) is a less significant contribution.

**Other comments**
- The paper is missing a citation for a very similar related work: "Semantic Abstraction: Open-World 3D Scene Understanding from 2D Vision-Language Models", H. Ha, et al., CoRL 2022. Further comments in "questions for rebuttal" below.

**Quality Of The Limitations Section:**

Limitations are addressed clearly

**Questions For Rebuttal:**

- Add LERF preliminaries, and introduce + explain the method more clearly
- The term "object part query" is never properly defined or introduced. I found this term vague and hard to understand (i.e., “*conditions the object part query over this mask”*)
- Clarifying details on the method:
    - What does the “flood-fill” do?
    - How are you able to render an “object-centric point cloud” without the mask already available? Is this a crop of the scene near the 3D coordinate with the highest LERF activation?
    - You’re deciding which points belong to the same object by thresholding how similar the DINO features are for the other nearby points in this crop?
- Details on "robot capture", "scene reconstruction", and "NeRF regularization" can be reduced significantly, with the rest moving to the appendix.
- Another relevant baseline is "Semantic Abstraction" (mentioned above). They also compute multi-scale relevancy maps using CLIP, and they have the added advantage of using a scene encoder so they can operate with new scenes. It's not necessarily critical to compare to this, although that would be great, but the paper should least comment on the similarities and how the two works relate.

**Robotics Focus:**

Sufficient demonstration on hardware

**Summary Of Paper:**

This paper aims to enable a robot to grasp a specific object at a specific part, where the “object + part” combination is specified in natural language. The main idea is to use a neural radiance field, which has been trained to render CLIP features (the recently proposed LERF), to represent the 3D scene, and use LERF to enable part-specific grasping. The main contributions are an implementation of the LERF-based pipeline for part-specific grasping, showing how to link LERF relevancy maps with generated 6-DoF grasps, and proposing a two-stage pipeline for using LERF to output relevancy maps that are faithful to the specified object parts. The method is evaluated in the real world in terms of the ability to identify the correct object, the correct part, and to execute a successful grasp at this part. The results show the proposed method outperforms one using a similar recently proposed scene representation and enables physically successful grasping 69% of the time.

**Summary Of Recommendation:**

The core capability that's developed and demonstrated has definite utility, the pipeline shows a robotics use case for an interesting and powerful new tool (LERF), and the real-world demonstrations are convincing. However, the paper is both framed too broadly ("task-oriented" rather than "part-oriented" grasping) and the key technical components are not communicated clearly.

---

Updating my score to weak accept after the rebuttal period

---

### Official Review · Reviewer_5Sor · 2023-07-20

**Confidence:** 4
**Originality:** Very Good
**Technical Quality:** Excellent
**Clarity Of Presentation:** Very Good
**Impact:** 4

**Recommendation:**

Strong Accept: I recommend accepting the paper and will argue for my recommendation even if other reviewers hold a different opinion.

**Review:**

**Strengths:**
- The experiments clearly demonstrate successful execution on the real robot with significantly high rates.
- The paper presents a way to perform task-oriented grasping while utilizing knowledge of general (foundational) models trained for grasping, vision-language understanding, and segmentation.
- The proposed approach does not use fine-tuning or demonstrations to learn grasps or affordances.
- The paper is well written, covering all the details necessary to understand the proposed pipeline.

**Weakness or areas of improvement:**
- It is unclear whether the authors employ the Evo-NeRF’s early stop training method.
- The authors show that they can deal with a scenario where the scene has multiple mugs if the description specifies the unique properties of the mug of interest. It is unclear if the method can handle cases where the description still fails to disambiguate a few instances of the object in the scene.

**Quality Of The Limitations Section:**

Limitations are addressed clearly

**Questions For Rebuttal:**

Please refer to the comments in the "Weakness" section.

**Robotics Focus:**

Sufficient demonstration on hardware

**Summary Of Paper:**

This work presents a pipeline LERF-TOGO that leverages LERF (Language Embedded Radiance Fields) alongside DINO-informed segmentation to perform semantically aware grasping with the help of language instructions. Given the description of the object of interest, LERF is used to identify where that object lies in the scene. A 3D object mask is segmented using flood fill starting from the point of interest identified by LERF. The flood fill is thresholded based on DINO features. Given the segmented object of interest, a language description of the part of interest is provided and used to further identify the portion of the segmented object that is of interest. GraspNet is used to derive grasps given multiple view angles, and proposed grasps are then weighted based on their geometric validity and semantic relevance. The results show that the proposed approach selects the grasps on the correct parts of the object of interest.

**Summary Of Recommendation:**

I believe the work would be valuable to the community. I would like the issues raised to be addressed in the rebuttal phase.

**Post-rebuttal update:** The authors have answered my questions. I appreciate the improvement in the quality of the manuscript. My recommendation to accept this paper remains.

---

### Official Review · Reviewer_B1ts · 2023-08-01

**Confidence:** 4
**Originality:** Good
**Technical Quality:** Very Good
**Clarity Of Presentation:** Very Good
**Impact:** 3

**Recommendation:**

Weak Reject: I recommend rejecting the paper, but will not argue for my recommendation if the majority of other reviewers have a different opinion.

**Review:**

The paper extends the ideas presented in LERF and tackles some of issues in it, especially with respect to part based grasping. The paper is of good technical quality with clear motivation and rationale for certain design decisions. I also found the presentation of the paper quite convincing with sufficient clairty in figures and writing. My only concern is the authors could have added further studies on real world robotics like static scene assumption, restrictions of camera capture etc. which can help in a compelling contribution.

**Strengths:**

- Utilizing LeRF for grasp selection is a good decision due to their strong zero shot performance and usefulness for open ended queris via pretraind model embeddings.

- Using DINO features for constructing Object masks overcomes the LeRF limitation of missing objectness

- Using 3D object masks for grasp selection (ranking) helps in grasping appropriate object part given in the query


**Weakness:**
- To construct a LeRF, multiview images of the object scene are needed. The authors have shown experiments on an arm robot with camera mounted on the on top of the gripper. Dealing with robots such as Fetch whose camera is fixed to the torso will be challenging task.

- Train LeRF needs some time. Here it is stated that it takes around 30 and 70 seconds for image capturing and training respectively. A faster way to train LeRF can to be checked for (near) real-time sensitive applications (Note: I acknowledge that its mentioned in the limitations but some experiment on this can help the paper?)

- It is assumed that the object query part will be given in the query. Giving the appropriate object part to grasp every time needs to be addressed.

**Quality Of The Limitations Section:**

Limitations are addressed clearly

**Questions For Rebuttal:**

- What would happen if the object scene changes? Can we adapt the model to deal with a simple task, for example clearing out a set of objects on table? Basically how can the model deal with such fixed (and perhaps known) changes from the recorded scene

- An ablation / possible observations on (1) effect of LeRF training steps could be shown to tell its effects ; (2) on the proportion of two scores used in calculating the s score for grasp sampling.

- Here a hemisheprical region is used to get scene images. What would be the impact of reducing the area to half of what is used? This would be interesting to see on a robots where the image capture area might be limited (e.g. Fetch robot where the camera is attached to the robot)

**Robotics Focus:**

Sufficient demonstration on hardware

**Summary Of Paper:**

This paper focuses on task specific grasp generation by leveraging the prior work of Language Embedded Radiance Fields (LERF) which proposed grounding language embeddings into a NeRF scene for open ended language queries onto it. The zero shot performance of LERF is used to conditionally query for a relevancy mask on an extracted object's point cloud from the overall scene. This mask then guides the task based grasping by focusing grasp generation on an object's region of interest. The usage of LERF also allows for text based queries to be sent as a high level command to the robot giving an accesible natural language interface.

**Summary Of Recommendation:**

This is nicely presented paper with clear details and motivation. However I feel it to be a minor extension to the key ideas presented in LERF as the major change seems to be in the spatial grouping within the scene. The idea is definitely worth exploring and the results seem to be encouraging so its not bad in that aspect but perhaps some additional contributions, perhaps along real world robotics aspects could help readers appreciate it even more.

---

### Official Review · Reviewer_bAJo · 2023-08-01

**Confidence:** 3
**Originality:** Good
**Technical Quality:** Excellent
**Clarity Of Presentation:** Excellent
**Impact:** 4

**Recommendation:**

Strong Accept: I recommend accepting the paper and will argue for my recommendation even if other reviewers hold a different opinion.

**Review:**

Strengths:
- The importance of task-oriented grasping is well-motivated and the authors propose a simple and elegant approach
- The approach requires no dataset collection, instead leveraging the knowledge contained in large pretrained visual language models
- The authors run extensive experiments on a real-robot with 31 real objects and show convincing qualitative results
- In addition to quantitative results, the authors also run ablations of the system and present analysis of the failure modes which I find interesting and informative

Most of the main weaknesses I see have all been acknowledged by the authors in the limitations section, namely:
- Training a LERF for each new scene takes on the order of minutes per grasp, which limits the practicality of the approach until the training time can be cut down significantly
- Even if LERF training is sped up significantly, having to take 60 new images with the robot arm moving to various angles for each new grasp also takes a long time, and is difficult to speed up unless additional cameras are placed in the scene, which would also limit the practicality of the approach
- The authors argue that previous methods which collect datasets for specific object don't scale to novel objects. However, the proposed approach still requires a human to annotate the correct part to grasp for each novel object. Perhaps an LLM might be able to propose parts but it would need to be shown that LLMs have sufficient understanding of this type of object manipulation.

**Quality Of The Limitations Section:**

Limitations are addressed clearly

**Questions For Rebuttal:**

While the evaluation of the system is thorough, I am wondering whether it would be possible to have more baseline comparisons with existing methods. As of now, the main baseline comparison to a different method is with ConceptFusion, which does not have physical evaluation.

Minor comments:
- Figure 2: Swap pictures of screwdriver and cleaning spray on left side to match the right side
- Table 1: Swap rows for part and object to match ordering in the text
- Table 1 says 82%, should it be 81%?


**Robotics Focus:**

Sufficient demonstration on hardware

**Summary Of Paper:**

This paper studies task-oriented grasping, in which objects must be grasped by a specific part (to avoid, e.g., damaging the object). The approach used builds on top of Language Embedded Radiance Fields (LERFs). For a given scene, the approach first constructs a LERF of the scene. Then, DINO features are used to generate a 3D mask for the object. Grasps are then sampled by quering LERF within this mask. The 3D mask helps constrain the LERF outputs as it has no concept of objectness. The authors test their approach on a set of 31 physical objects, and find that 81% of generated grasps are on the correct object part.

**Summary Of Recommendation:**

This paper proposes a simple and elegant approach for task-oriented grasping. The main contribution is on the systems level, combining various techniques (such as LERF, DINO, GraspNet, etc.) to build a robotic system and show its effectiveness. While there are limitations of the approach, I find the results quite interesting and would be of value to the community. Overall I am enthusiastic about this paper.

---

### Decision · Program_Chairs · 2023-08-30

**Decision:**

Accept (Oral)

**Comment:**

The paper introduces a novel method for task-oriented grasping by reconstructing a Language Embedded Radiance Field.

After the rebuttal, the reviewers achieve consensus on accepting the paper.
Majority concerns of the paper have been addressed.

Based on the scores of the reviewers and the novelty of the paper, I recommend to accept the paper as an oral presentation.